# SHIFTADDNAS: HARDWARE-INSPIRED SEARCH FOR MORE ACCURATE AND EFFICIENT NEURAL NETWORKS

## ABSTRACT

Neural networks (NNs) with intensive multiplications (e.g., convolutions and transformers) are powerful yet power hungry, impeding their more extensive deployment into resource-constrained edge devices. As such, multiplication-free networks, which follow a common practice in energy-efficient hardware implementation to parameterize NNs with more efficient operators (e.g., bitwise shifts and additions), have gained growing attention. However, multiplication-free networks in general under-perform their vanilla counterparts in terms of the achieved accuracy. To this end, this work advocates hybrid NNs that consist of both powerful yet costly multiplications and efficient yet less powerful operators for marrying the best of both worlds, and proposes **ShiftAddNAS**, which can automatically search for more accurate and more efficient NNs. Our ShiftAddNAS highlights two enablers. Specifically, it integrates (1) the first hybrid search space that incorporates both multiplication-based and multiplication-free operators for facilitating the development of both accurate and efficient hybrid NNs; and (2) a novel weight sharing strategy that enables effective weight sharing among different operators that follow heterogeneous distributions (e.g., Gaussian for convolutions vs. Laplacian for add operators) and simultaneously leads to a largely reduced supernet size and much better searched networks. Extensive experiments and ablation studies on various models, datasets, and tasks consistently validate the effectiveness of ShiftAddNAS, e.g., achieving up to a **+7.7%** higher accuracy or a **+4.9** better BLEU score as compared to state-of-the-art expert-designed and neural architecture searched NNs, while leading to up to **93%** or **69%** energy and latency savings, respectively. All the codes will be released upon acceptance.

## 1 INTRODUCTION

The unprecedented performance achieved by neural networks (NNs), e.g., convolutional neural networks (CNNs) and Transformers, requires intensive multiplications and thus prohibitive training and inference costs, contradicting the explosive demand for embedding various intelligent functionalities into pervasive resource-constrained edge devices. In response, multiplication-free networks have been proposed to alleviate the prohibitive resource requirements by replacing the costly multiplications with lower-cost operators for boosting hardware efficiency. For example, AdderNet (Chen et al., 2020) utilizes mere additions to design NNs; and ShiftAddNet (You et al., 2020a) follows a commonly used hardware practice to re-parameterize NNs with both bitwise shifts and additions. Despite their promising performance in hardware efficiency, multiplication-free NNs in general under-perform their CNN and Transformer counterparts in terms of task accuracy for both computer vision (CV) and natural language processing (NLP) applications.

To marry the best of both worlds, we advocate hybrid multiplication-reduced network architectures that integrate both multiplication-based operators (e.g., vanilla convolution (Krizhevsky et al., 2012) and attention (Vaswani et al., 2017)) and multiplication-free operators (e.g., shift and add (You et al., 2020a)) to simultaneously boost task accuracy and efficiency. Thanks to the amazing success of neural architecture search (NAS) in automating the process of designing state-of-the-art NNs, it is natural to consider NAS as the design engine of the aforementioned hybrid NNs for various applications and tasks, each often requiring a different performance-efficiency trade-off. However, there still exist a few challenges in leveraging NAS to design the hybrid NNs. *First*, existing NAS methods mostly consider the search for either efficient CNNs (Wan et al., 2020), Transformers (Chen et al., 2021b), or hybrid CNN-Transformers (Ding et al., 2021; Li et al., 2021), and there still is a

*lack of a seminal work that searches for multiplication-reduced hybrid networks, especially for the hardware-inspired networks that incorporate both bitwise shifts and additions. Second*, a hybrid search space could make it more challenging to achieve effective NAS and further aggravate the search burden, due to the enlarged search space imposed by the newly introduced multiplication-free operators. It is worth noting that *existing weight sharing strategies of NAS do not directly apply to the target hybrid search space, because weights of different operators follow heterogeneous distributions, leading to a dilemma of either inefficient search or inconsistent architecture ranking.* Specifically, weights in convolutional and adder layers follow Gaussian and Laplacian distributions, respectively, as also highlighted by (Chen et al., 2020; Xu et al., 2020). As such, forcing weight sharing among heterogeneous operators could hurt the capacity and thus the achieved accuracy of the resulting NNs, while treating them separately could explode the search space and make it more difficult to achieve effective NAS, i.e., the dilemma mentioned above.

To tackle the aforementioned challenges towards more accurate and efficient NNs, this work makes the following contributions:

1. We propose a generic NAS framework dubbed **ShiftAddNAS**, which for the first time can automatically search for efficient hybrid NNs with both superior accuracy and efficiency. Our ShiftAddNAS integrates a hybrid hardware-inspired search space that incorporates both multiplication-based operators (e.g., convolution and attention) and multiplication-free operators (e.g., shift and add), and can serve as a play-and-plug module to be applied on top of SOTA NAS works for further boosting their achievable accuracy and efficiency.

2. We develop a new weight sharing strategy for effective search with hybrid search spaces, which only incurs a negligible overhead when searching for hybrid operators with heterogeneous (e.g., Gaussian vs. Laplacian) weight distributions as compared to the vanilla NAS with merely multiplication-based operators, alleviating the dilemma mentioned above regarding either inefficient search or inconsistent architecture ranking.

3. We conduct extensive experiments and ablation studies to validate the effectiveness of ShiftAddNAS against state-of-the-art (SOTA) works. Results on multiple benchmarks demonstrate the superior accuracy and hardware efficiency of its searched NNs as compared to both (1) manually designed multiplication-free networks, CNNs, Transformers, and hybrid CNN-Transformers, and (2) SOTA NAS works, on both CV and NLP tasks.

## 2 RELATED WORKS

**Multiplication-free NNs.** Many efficient NNs aim to reduce their intensive multiplications that dominate the time/energy costs. One important trend is to replace the multiplications with lower-cost operators: BNNs (Courbariaux et al., 2016; Juefei-Xu et al., 2017) binarize both the weights and activations and reduce multiplications to merely sign flips at non-negligible accuracy drops; AdderNets (Chen et al., 2020; Xu et al., 2020; Wang et al., 2021b) fully replace the multiplications with lower-cost additions and further develop an effective backpropagation scheme for efficient AdderNet training; Shift-based NNs leverage either spatial shift (Wu et al., 2018) or bit-wise shift operations, e.g., DeepShift (Elhoushi et al., 2021), to reduce the amount of multiplications; and ShiftAddNet (You et al., 2020a) draws inspirations from efficient hardware designs to reparamatize NNs with mere bitwise shifts and additions. While multiplication-free NNs under-perform their vanilla NN counterparts in terms of achieved accuracy, ShiftAddNAS aims to automatically search for multiplication-reduced NNs that incorporate both multiplication-based and multiplication-free operators for marrying the best of both worlds, i.e., boosted accuracy and efficiency.

**Neural architecture search.** NAS has achieved an amazing success in automating the design of efficient NN architectures. For searching for CNNs, early works (Tan & Le, 2019; Tan et al., 2019; Howard et al., 2019) adopt reinforcement learning based methods that require a prohibitive search time and computing resources, while recent works (Liu et al., 2018; Wu et al., 2019a; Wan et al., 2020; Yang et al., 2021) utilize differentiable search to greatly improve the search efficiency. More recently, SOTA works adopt one-shot NAS (Guo et al., 2020; Cai et al., 2019; Yu et al., 2020; Wang et al., 2021a) to decouple the architecture search from supernet training and then evaluates the performance of sub-networks whose weights are directly inherited from the pretrained supernet. For searching for Transformers, recently emerging works (Wang et al., 2020a; Su et al., 2021; Chen et al., 2021b;a) adopt one-shot NAS and an evolutionary algorithm to search for optimal Transformer architectures for both NLP and CV tasks. Additionally, BossNAS (Li et al., 2021) and HR-NAS (Ding et al., 2021) further search for hybrid CNN-Transformer architectures.

Nevertheless, little effort has been made to exploring NAS methods especially their search strategies for multiplication-reduced NNs that incorporate both multiplication-based and multiplication-free operations. Furthermore, it is not clear whether existing efficient NAS methods are applicable to searching for such multiplication-reduced NNs. Specifically, prior weight sharing strategies may not work since weights and activations in CNNs and AdderNets follow a different distribution (Chen et al., 2020). As such, it is highly desirable to develop NAS methods, e.g., ShiftAddNAS, dedicated for hardware-inspired multiplication-reduced NNs to increase achievable accuracy and efficiency.

**Transformers.** Transformers (Vaswani et al., 2017) were first proposed for NLP tasks, which has inspired many interesting works. Some advance Transformer design by improving the attention mechanism (Chen et al., 2018), training deeper Transformers (Wang et al., 2019), and replacing the attention with convolutional modules (Wu et al., 2019b); and others strive to reduce Transformers' computational complexity by adopting sparse attention mechanisms (Zaheer et al., 2020), low-rank approximation (Wang et al., 2020b), or compression techniques (Wu et al., 2020). Recently, there has been a growing interest in developing Transformers for CV tasks: Vision Transformer (ViT) (Dosovitskiy et al., 2021) for the first time successfully applies pure Transformers to image classification and achieves SOTA task accuracy, which yet relies on pretraining on giant datasets (Hinton et al., 2015); following works including DeiT (Touvron et al., 2021) T2T-ViT (Yuan et al., 2021) develop new training recipes and tokenization schemes, for achieving comparable accuracy without the necessity of costly pretraining; and another trend is to incorporate CNN modules into Transformer architectures for better accuracy and efficiency tradeoffs (Wu et al., 2021; Xiao et al., 2021; Graham et al., 2021). In contrast, we advocate hybrid multiplication-reduced NNs and develop an automated search framework that can automatically search for such hardware inspired hybrid models.

## 3 THE PROPOSED SHIFTADDNAS FRAMEWORK

In this section, we first introduce the hybrid search space from both algorithmic and hardware costs perspectives, providing high-level background and justification for motivating ShiftAddNAS; Sec. 3.2 elaborates the one-shot search method of ShiftAddNAS by first analyzing the dilemma of either inefficient search or inconsistent architecture ranking and then introducing the proposed novel heterogeneous weight sharing strategy tackling the aforementioned dilemma.

### 3.1 SHIFTADDNAS: HYBRID SEARCH SPACE

**Candidate blocks.** The first step of developing ShiftAddNAS is to construct a hybrid search space incorporating suitable building blocks that exhibit various performance-efficiency trade-offs. Specifically, we hypothesize that integrating both multiplication-based and multiplication-free blocks into the search space could lead to both boosted accuracy and efficiency, and consider blocks from two trends of designing NNs: (1) *capable* NNs, e.g., vanilla CNNs and Transformers, leverage either convolutions (`Conv`) or multi-head self-attentions (`Attn`) that comprise of intensive multiplications to capture local or global correlations, achieving a SOTA accuracy in both CV and NLP tasks; and (2) *efficient* multiplication-free NNs, e.g, ShiftAddNet, draw inspirations from hardware design practices to incorporate two efficient and complementary blocks, i.e., coarse-grained `Shift` and fine-grained `Add`, for favoring hardware efficiency, while maintaining a decent accuracy. While our constructed general hybrid search space for both NLP and CV tasks are shown in Fig. 2, we next analyze the building blocks from both algorithmic and hardware costs perspectives.

- `Attn` is a core component of Transformers (Vaswani et al., 2017), which consists of a number of heads H with each capturing different global-context information by measuring pairwise correlations among tokens as defined below:

$$\boldsymbol{O}_{\textbf{Attn}} = \texttt{Concat}(\text{H}_1, \cdots, \text{H}_h) \cdot W^O, \text{ where } \text{H}_i = \texttt{Softmax}\left(\frac{QW_i^Q \cdot (KW_i^K)^T}{\sqrt{d_k}}\right) \cdot VW_i^V, \quad (1)$$

  where $h$ denotes the number of heads, $Q, K, V \in \mathbb{R}^{n \times d}$ are the query, key, and value embeddings of hidden dimension $d$ obtained by linearly projecting the input sequence of length $n$. For each head, $W_i^Q, W_i^K, W_i^V \in \mathbb{R}^{d \times d_k}$ are learned projection weight matrices where $d_k = d/h$ is the embedding dimension of each head. In this way, the `Attn` block first computes dot-products between key-query pairs, scales to stabilize the training, uses `Softmax` to normalize the resulting attention scores, and then computes a weighted sum of the value embeddings corresponding to different inputs. Finally, the results from all heads are concatenated and further projected with a weight matrix $W^O \in \mathbb{R}^{d \times d}$ to generate the outputs.

- **Conv** is a key operator of CNNs, which models the local-context information of high-dimensional inputs such as images through sliding kernel weights $W$ on top of inputs $X$ to measure their similarity (Gu et al., 2018), as defined in Eq. (2). Its translation invariant and weight sharing ability leads to various SOTA CNNs (He et al., 2016) or hybrid CNN-Transformer models (Xiao et al., 2021). However, the computational complexities of CNNs can be prohibitive due to their required intensive multiplications. For example, one forward pass of ResNet-50 (He et al., 2016) requires 4G floating point multiplications.

$$\boldsymbol{O}_{\mathbf{Conv}} = \sum X^T * W; \ \boldsymbol{O}_{\mathbf{Shift}} = \sum X^T * (S \cdot 2^P); \ \boldsymbol{O}_{\mathbf{Add}} = -\sum \|X - W\|_1, \quad (2)$$

- **Shift** is a well-known efficient hardware primitive, motivating the recent development of shift-based efficient NNs. For example, DeepShift (Elhoushi et al., 2021) parametrizes NNs with bit-wise shifts and sign flips, as formulated in the middle of Eq. (2), with $W = S \cdot 2^P$ denoting weights in the shift blocks, where $S \in \{-1, 0, 1\}$ are sign flip operators and the power-of-two parameter for $P$ represents the bitwise shifts. However, NNs built with shift blocks and quantized weights are observed to be inferior to multiplication-based NNs in terms of expressiveness (accuracy) as validated in (You et al., 2020a).

- **Add** is another efficient hardware primitive which motivates recent works (Chen et al., 2020; Wang et al., 2021b; Song et al., 2021) to design efficient NNs using merely additions to measure the similarity between kernel weights $W$ and inputs $X$, as shown in the right part of Eq. (2). Such add-based NNs (Chen et al., 2020; Xu et al., 2020) in general have a better expressive capacity than their shift-based counterparts. For example, AdderNets (Chen et al., 2020) achieve a 1.37% higher accuracy than DeepShift under similar or even lower FLOPs on ResNet-18 with the ImageNet dataset. However, add-based operators (i.e., repeated additions) are not parameter-efficient as compared to bitwise shift operations (You et al., 2020a). While NNs combining shfit and add achieve a boosted accuracy, efficiency, and robustness than NNs using merely either of them, their accuracy still compares unfavorably as compared with vanilla CNNs or Transformers.

Based on the above introduction, the search space in ShiftAddNAS incorporates all the four different types of blocks (i.e., Attn, Conv, Shift, and Add), aiming to push forward both NNs' accuracy and efficiency. Note that we refer to all operators as blocks, and adopt block based search space because it has been evidenced and proven that block based ones can reduce the search space size and lead to more accurate architecture ranking/rating (Li et al., 2020b;a).

**Hardware cost analysis.** As mentioned, multiplication-based operators (e.g., **Attn** and **Conv**) favor a superior accuracy yet is not hardware efficient, while multiplication-free operators (e.g., **Shift** and **Add**) favors a better hardware efficiency yet can hurt the achievable accuracy. Specifically, as shown in Fig. 1, bitwise shifts can save as high as 196× and 24× energy costs over multiplications, when implemented in a 45nm CMOS technology and SOTA FPGA (Xilinx Inc.), respectively; with a 16-bit precision, bitwise shifts may achieve at least 9.7× and 1.45× average power and area savings than multipliers (Elhoushi et al., 2021); and similarly, additions can save up to 196× and 31× energy costs over multiplications in 32-bit fixed-point (FIX32) formats, and 47× and 4.1× energy costs in 32-bit floating-point (FP32) formats,

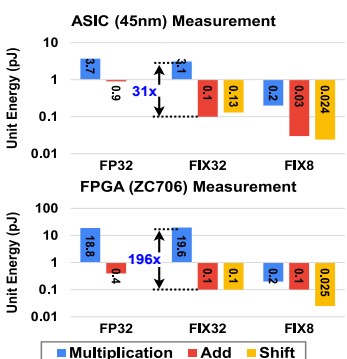

Figure 1: Unit energy comparisons.

when implemented in a 45nm CMOS technology and SOTA FPGA (Xilinx Inc.), respectively, while aggressively leading to 1.84×, 25.5×, and 7.83× area savings than multiplications in a 45nm CMOS technology with FP32, FIX32, and FIX8 formats, respectively (Chen et al., 2021c).

**Supernet for NLP tasks.** Based on the above search space, we construct a supernet for the convenience of search following SOTA one-shot NAS methods (Cai et al., 2018; Guo et al., 2020) by estimating the performance of each candidate hybrid model (i.e., subnet) without fully training it. As shown in Fig. 2 (a), each macro-block in the supernet includes all the aforementioned four candidate blocks and three multi-branch combinations (e.g., Attn+Conv) along the channel dimension for capturing both global and local context information following (Wu et al., 2020), where the candidate blocks in the same layer are isolated with each followed by two-layer MLPs and enabling elastic embedding dimension, head numbers, and MLP hidden dimension for fine-grained search for efficient NNs as (Wang et al., 2020a). Overall, our supernet for NLP tasks contains about $10^{14}$ subnet

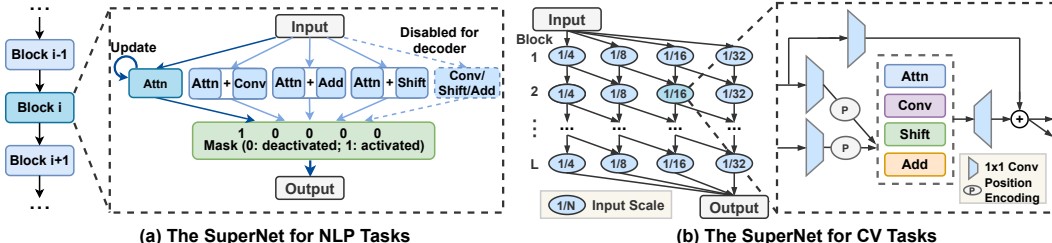

**Figure 2:** Supernets for NLP and CV tasks: (a) For NLP, we adopt a multi-branch structure for each block of the supernet, where Attn+Conv represents the channel-wise concatenation of these two blocks, and (b) for CV tasks, we consider a multi-resolution pipeline for each block of the supernet.

candidates, and the searchable choices are listed in Tab. 1. During training, all possible subnets are uniformly sampled and only one path is activated for each layer at run-time considering the practical concern on memory consumption for supernet training. For ease of evaluation, we incorporate common treatments of NAS in our suppnet design. First, for the elastic dimensions mentioned above, all sub-

Table 1: The search space for NLP tasks.

| | |
|---|---|
| Encoder block types | [Attn, Attn+Conv, Attn+Shift] [Attn+Add, Conv, Shift, Add] |
| Decoder block types | [Attn, Attn+Conv] [Attn+Shift, Attn+Add] |
| Num. of decoder blocks | [6, 5, 4, 3, 2, 1] |
| Elastic embed. dim. | [1024, 768, 512] |
| Elastic head num. | [16, 8, 4] |
| Elastic MLP dim. | [4096, 3072, 2048, 1024] |
| Arbitrary Attn | [3, 2, 1] |

nets share the front portion of weights or heads of the largest dimension. Second, all decoder blocks can take the last one, two, or three encoder blocks as inputs for abstracting both high-level and low-level information (Wang et al., 2020a). Note that the number of decoder blocks are also searchable and the conv, shift and add operators are disabled for decoder blocks, as they are observed to be sensitive and activating those paths might hurt the accuracy (You et al., 2020a; Wu et al., 2019b).

**Supernet for CV tasks.** Different from the commonly used elastic hidden dimension design for NLP tasks, various spatial resolutions or scales are essential for CV tasks. As such, to ensure more capable feature description of the searched NNs, we adopt a multi-resolution supernet design. As

Table 2: The search space for CV tasks.

| | |
|---|---|
| Block types | [Attn, Conv, Shift, Add] |
| Num. of $56^2 \times 128$ blocks | [1, 2, 3, 4] |
| Num. of $28^2 \times 256$ blocks | [1, 2, 3, 4] |
| Num. of $14^2 \times 512$ blocks | [3, 4, 5, 6, 7] |
| Num. of $7^2 \times 1024$ blocks | [4, 5, 6, 7, 8, 9] |

shown in Fig. 2 (b), the supernet incorporates flexible downsampling options, where the spatial resolution for each layer can either stay unchanged or be reduced to half of its previous layer's scale until reaching the smallest resolution. In this way, the four candidate blocks can work collaboratively to deliver the multiscale features required by most CV tasks. Overall, our supernet contains about $10^9$ subnets, for which the detailed searchable choices are summarized in Tab. 2. Note that the Attn block is followed by two-layer MLPs and we also include a residual connection for each block as inspired by (Srinivas et al., 2021). During training, the supernet performs uniform sampling and only activates one path of the chosen resolution and block type for each layer as for the NLP tasks.

## 3.2 SHIFTADDNAS: SEARCH METHOD

### 3.2.1 BACKGROUND AND FORMULATION OF ONE-SHOT NAS

We adopt one-shot NAS for improved search efficiency, i.e., assuming that the subnet candidates can directly inherit their weights from the supernet, following SOTA NAS works. Such a strategy is commonly referred as *weight sharing*. Specifically, the supernet $\mathcal{N}$ with parameters $\mathbf{W}$ is trained to obtain the weights for all subnets within the search space $\mathcal{S}$. Since the supernet training and architecture search are decoupled in one-shot NAS, it usually requires two-level optimization: supernet training and architecture evaluation as defined below:

$$\mathbf{W}_{\mathcal{S}} = \arg\min_{W} L_{train}(\mathcal{N}(\mathcal{S}, W)), \tag{3}$$

$$a^* = \arg\max_{a \in \mathcal{S}} ACC_{val}(\mathcal{N}(a, \mathbf{W}_{\mathcal{S}}(a))). \tag{4}$$

where $\mathcal{N}(\mathcal{S}, \mathbf{W})$ represents all possible candidate subnets within the search space. We first train the supernet by uniformly sampling different subnets $a$ from $S$ as formulated in Eq. (3), after which all subnet candidates $a$ directly inherit their corresponding weights $\mathbf{W}_{\mathcal{S}}(a)$ from the supernet $\mathbf{W}_{\mathcal{S}}$. Finally, we evaluate the accuracy $ACC_{val}(.)$ of each path on the validation set and search for the best subnet with the highest accuracy as formulated in Eq. (4).

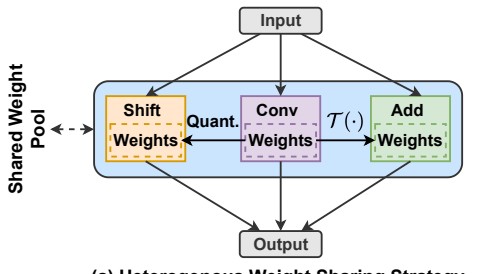
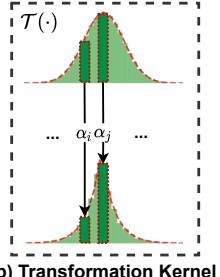

**(a) Heterogenous Weight Sharing Strategy**  **(b) Transformation Kernel**

Figure 4: (a) Illustration of the proposed heterogeneous weight sharing strategy, where weights of shift blocks are quantized to powers of two; (b) visualization of the adopted learnable transformation kernel $\mathcal{T}(\cdot)$ for mapping the shared weights of Gaussian distribution to a Laplacian distribution.

### 3.2.2 PROPOSED HETEROGENEOUS WEIGHT SHARING STRATEGY

**Dilemma of vanilla ShiftAddNAS.** The target hybrid search space of ShiftAddNAS inevitably enlarges the supernet due to the newly considered operators. As such, activating all block choices without weight sharing as (Gong et al., 2019; Cai et al., 2018) can easily explode the memory consumption of NAS. On the other hand, directly sharing weights among different operators as (Chen et al., 2021b) will lead to biased search,

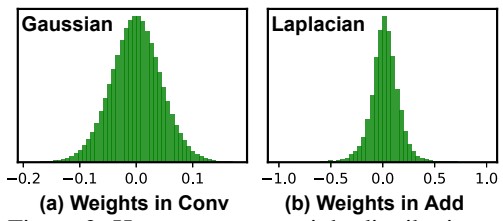

**(a) Weights in Conv**   **(b) Weights in Add**
Figure 3: Heterogeneous weight distributions.

especially for our hardware-inspired hybrid search space where weights and activations of different operators follow heterogeneous distributions, e.g., weights of the Conv and Add blocks follow a Gaussian and Laplacian distribution, respectively, as shown in Fig. 3 and also highlighted in (Chen et al., 2020). Specifically, if we follow the existing weight sharing strategy to enforce a homogeneous weight distribution among different operators during training the supernet, the resulting weights will not match the heterogeneous weight distributions of independently trained optimal hybrid subnets. That is to say, for NAS with the target hybrid search space, there exists an optimization gap between the goals of weight sharing optimization and individual subnet optimization, where the former is approximated while the latter is accurate (Xie et al., 2020). Hence, naively adopting the homogeneous weight sharing strategy can lead to inconsistent architecture ranking, which is a major issue associated with one-shot NAS as pointed out by (Chu et al., 2019; You et al., 2020b).

**Proposed solution: heterogeneous weight sharing.** To tackle the aforementioned dilemma, we propose a heterogeneous weight sharing strategy that can simultaneously reduce the supernet size corresponding to the target hybrid search space and allow weights of different blocks to follow heterogeneous distributions. Specifically, the learning objective for the supernet includes both the traditional cross-entropy loss and a KL-divergence loss that is to regularize weight distributions to be close to either a standard Gaussian distribution $\mathcal{N}(0, I)$ or Laplacian distribution $\mathcal{L}_p(0, \lambda)$, (Xie et al., 2020; Chen et al., 2020), where $I$ is the identity matrix and $\lambda = 1$, dedicated for the Conv and Add blocks, respectively, to reduce the aforementioned optimization gap as formulated in Eq. (5):

$$
\begin{aligned}
\mathcal{L}_{\mathcal{S}} = \mathcal{L}_{CE} + \mathcal{L}_{KL} = &-\frac{1}{N}\sum_{i=1}^{N} P(y_i|x_i)\log(P(\hat{y}_i|x_i)) \\
&+ \mathcal{D}_{KL}(P_{\texttt{Conv}}(\mathbf{W}_{\mathcal{S}}) \, || \, \mathcal{N}(0, I)) + \mathcal{D}_{KL}(P_{\texttt{Add}}(\mathcal{T}(\mathbf{W}_{\mathcal{S}})) \, || \, \mathcal{L}_p(0, \lambda)),
\end{aligned}
\tag{5}
$$

where $\{(x_i, y_i)\}_{i=1}^{N}$ are training data, $\hat{y}$ denotes the output prediction, and $\mathcal{D}_{KL}(p||q) = -\int p(z)\frac{p(z)}{q(z)}dz$ measures the KL-divergence between two distributions. During training, we maintain a shared weight pool for each layer to share weights across all the **Conv**, **Add**, and **Shift** blocks, as illustrated in Fig. 4 (a). Meanwhile, weights of the **Conv** blocks directly leverage the corresponding ones in the shared weight pool for both forward and backpropagation, while being encouraged to follow a Gaussian distribution by the objective function; weights of the **Shift** blocks quantize the shared weights of Gaussian distribution to powers of two before multiplying with the input features; and for the **Add** blocks, we make use of **a learnable transformation kernel** $\mathcal{T}(\cdot)$ to map the shared weights of Gaussian distribution to a Laplacian distribution. For the learnable transformation kernel as captured by Eq. (6), the core idea is to apply a piece-wise linear transformation after flattening and sorting the weights in a descending order, and then to reshape and rearrange the transformed weights back to their positions before sorting.

$$\mathcal{T}(W) = \sum_{i=0}^{d-1} \alpha_i \times W_{[s \times i : s \times (i+1)]}, \tag{6}$$

where $\{\alpha_i\}_{i=1}^{d}$ denote the learnable parameters in $\mathcal{T}(\cdot)$, $\{W_i\}_{i=1}^{n}$ represent the sorted weights (a total of $n$) in the pool, $s = n/d$ denotes an interval within which the transformation is linear, as illustrated in Fig. 4 (b). As validated in our experiment (e.g., Fig. 3), such a transformation kernel can successfully transform the shared weights of Gaussian to the desired Laplacian distribution, which is consistent with previous observations about kernel learning via linear transformation (Jain et al., 2012). In our design, each layer has its own learnable kernel $\mathcal{T}(\cdot)$ with a dimension $d$ of 200 throughout all the experiments as we observed that such a dimension is adequate to learn the transformation across all the models and datasets, leading to over 40% supernet size reduction while only incurring a negligible ($< 0.01\%$ of the supernet size and computational costs) search overhead. After the supernet is well trained, evolution search is applied to find the optimal subnets.

## 4 EXPERIMENT RESULTS

In this section, we first describe our experiment setups, and then benchmark ShiftAddNAS over SOTA CNNs, Transformers, and previous NAS frameworks on both NLP and CV tasks. After that, we conduct ablation studies regarding ShiftAddNAS's heterogeneous weight sharing strategy.

### 4.1 EXPERIMENT SETUPS

**Datasets, baselines, and evaluation metrics.** *For NLP tasks,* we consider two machine translation datasets, WMT'14 English to French (En-Fr) and English to German (En-De), which consist of 36.3M and 4.5M pairs of training sentences, respectively. The train/val/test splits follow the tradition as in (Wang et al., 2020a; Gehring et al., 2017). We consider five baselines: Transformer (Vaswani et al., 2017), Lightweight Conv (Wu et al., 2019b), Lite Transformer (Wu et al., 2020), and two previous NAS works including Evolved Transformer (So et al., 2019) and HAT (Wang et al., 2020a). We evaluate in terms of five evaluation metrics: the number of parameters/FLOPs, achieved BLEU, and hardware energy and latency measured on a SOTA accelerator Eyeriss (Chen et al., 2016) clocked at 250MHz, where the BLEU is calculated with case-sensitive tokenization following (Wang et al., 2020a). *For CV tasks,* we consider the ImageNet dataset and four kinds of SOTA baselines: four multiplication-free CNNs (Chen et al., 2020; Xu et al., 2020; Courbariaux et al., 2016; Elhoushi et al., 2021), two CNNs (He et al., 2016; Hu et al., 2018), five Transformers (Dosovitskiy et al., 2021; Touvron et al., 2021; Yuan et al., 2021; Han et al., 2021; Srinivas et al., 2021), and four NAS works (i.e., HR-NAS (Ding et al., 2021), BossNAS (Li et al., 2021), AutoFormer (Chen et al., 2021b), and VITAS (Su et al., 2021)). Similar to those for the NLP tasks, we adopt five evaluation metrics: the number of parameters/MACs, achieved accuracy, and hardware energy and latency.

**Search and training settings.** *For NLP tasks,* after training the supernet for 40K steps, we adopt an evolutionary algorithm (Wang et al., 2020a) to search for subnets with various latency and FLOPs constraints ranging from 1.5G to 4.5G for 30 steps with a population of 125, a crossover population of 50, and a mutation population of 50 with a probability of 0.3. During search, measuring latency for each chosen subnet can be time-consuming. Instead, we estimate the latency using a three-layer NN trained with encoding architecture parameters as features and measured latency as labels following (Wang et al., 2020a). The latency predictor is accurate with an average prediction error of $< 5\%$. The searched subnets are then retrained from scratch for another 40K steps with an Adam optimizer and a cosine learning rate (LR) scheduler, where the LR is linearly warmed up from $10^{-7}$ to $10^{-3}$ and then annealed (same for training supernets). *For CV tasks,* we conduct an evolutionary search with FLOPs constraints for 20 steps with a population of 50, a crossover population of 25, and a mutation population of 25 with a probability of 0.2 following (Chen et al., 2021b). We train both the supernet and searched subnets using the same recipe and hyperparameters as DeiT (Touvron et al., 2021). Note that the position encoding in the attention blocks is replaced with a depthwise convolution following (Li et al., 2021) for reducing the computational complexity.

### 4.2 SHIFTADDNAS VS. SOTA ON NLP TASKS

We compare ShiftAddNAS with SOTA language models on two NLP tasks to evaluate its effectiveness. Fig. 5 shows that ShiftAddNAS consistently outperforms all the baselines in terms of BLEU scores and FLOPs. Specifically, ShiftAddNAS with full precision achieves **11.8% ∼ 73.6%** FLOPs reductions while offering a comparable or better BLEU score (-0.3 ∼ +1.1), over all the full precision baselines. To benchmark with Lite Transformer (8-bit) which is dedicated for mobile devices,

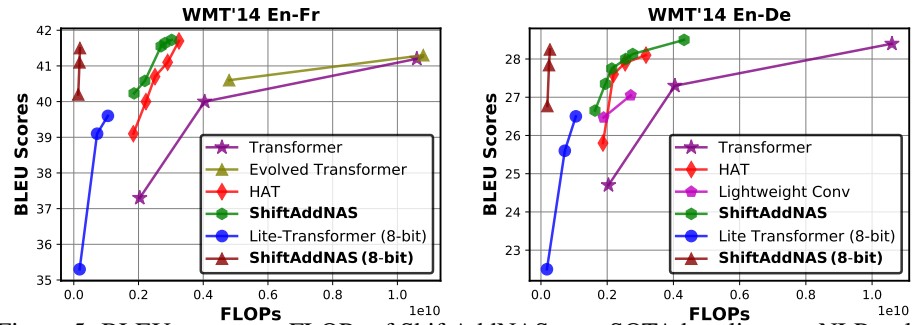

Figure 5: BLEU scores vs. FLOPs of ShiftAddNAS over SOTA baselines on NLP tasks.

Table 3: ShiftAddNAS vs. SOTA baselines in terms of accuracy and efficiency on NLP tasks.

| | WMT'14 En-Fr | | | | | WMT'14 En-De | | | | |
|---|---|---|---|---|---|---|---|---|---|---|
| | Params | FLOPs | BLEU | Latency | Energy | Params | FLOPs | BLEU | Latency | Energy |
| Transformer | 176M | 10.6G | 41.2 | 130ms | 214mJ | 176M | 10.6G | 28.4 | 130ms | 214mJ |
| Evolved Trans. | 175M | 10.8G | 41.3 | - | - | 47M | 2.9G | 28.2 | - | - |
| HAT | 48M | 3.4G | 41.4 | 49ms | 81mJ | 44M | 2.7G | 28.2 | 42ms | 69mJ |
| **ShiftAddNAS** | **46M** | **3.0G** | **41.8** | **43ms** | **71mJ** | **43M** | **2.7G** | **28.2** | **40ms** | **66mJ** |
| HAT | 46M | 2.9G | 41.1 | 42ms | 69mJ | 36M | 2.2G | 27.6 | 34ms | 56mJ |
| **ShiftAddNAS** | **41M** | **2.7G** | **41.6** | **39ms** | **64mJ** | **33M** | **2.1G** | **27.8** | **31ms** | **52mJ** |
| HAT | 30M | 1.8G | 39.1 | 29ms | 48mJ | 25M | **1.5G** | 25.8 | 24ms | 40mJ |
| **ShiftAddNAS** | **29M** | **1.8G** | **40.2** | **16ms** | **45mJ** | **25M** | 1.6G | **26.7** | **24ms** | **40mJ** |
| Lite Trans. (8-bit) | 17M | 1G | 39.6 | 19ms | 31mJ | 17M | 1G | 26.5 | 19ms | 31mJ |
| **ShiftAddNAS (8-bit)** | **11M** | **0.2G** | **41.5** | **11ms** | **16mJ** | **17M** | **0.3G** | **28.3** | **16ms** | **24mJ** |
| Lite Trans. (8-bit) | 12M | 0.7G | 39.1 | 14ms | 24mJ | 12M | 0.7G | 25.6 | 14ms | 24mJ |
| **ShiftAddNAS (8-bit)** | **10M** | **0.2G** | **41.1** | **10ms** | **15mJ** | **12M** | **0.2G** | **26.8** | **9.2ms** | **14mJ** |

we refer to a SOTA quantization technique (Banner et al., 2018) for quantizing ShiftAddNAS to 8-bit fixed point: ShiftAddNAS (8-bit) achieves **+1.8 ∼ +4.9** BLEU scores improvements over Lite Transformer (8-bit), while offering 5.0% ∼ 82.7% FLOPs reductions, and aggressively reduces **91.6% ∼ 98.4%** FLOPs as compared to all the full-precision baselines with comparable BLEU scores (-0.1 ∼ +0.3). Note that for quantized models, we follow (Zhou et al., 2016) to use FLOPs $\times (\text{Bit}/32)^2$ for calculating the effective FLOPs which is proportional to the number of bit-operations. We further compare various aspects of ShiftAddNAS with all baselines in Tab. 3. As illustrated in this Table, ShiftAddNAS consistently outperforms the baselines, e.g., achieves up to **+2** BLEU scores improvement when comparing ShiftAddNAS (8-bit) with Lite Transformer on WMT'14 En-Fr and **69.1%** and **69.2%** energy and latency savings when comparing ShiftAddNAS with Transformer on WMT'14 En-De, with a comparable or even fewer model parameters and FLOPs.

### 4.3 SHIFTADDNAS VS. SOTA ON CV TASKS

We further compare ShiftAddNAS over SOTA baselines on ImageNet to evaluate its effectiveness on the image classification task. As shown in Tab. 4, ShiftAddNAS outperforms a wide range of baselines. Here we refer MACs as Multiply–accumulate or Shift-accumulate operations. For example, ShiftAddNet-T0 (searched with a 4.5G MACs constraint) with 3.7G MACs achieves an improved top-1 accuracy of (1) **+5.3% ∼ +26.3%** over SOTA multiplication-free CNNs, (2) **+0.7% ∼ +6.0%** over SOTA CNNs, (3) **+0.4% ∼ +7.6%** over SOTA Transformers, (4) **+1.3% ∼ +4.8%** over SOTA CNN-Transformers, and (5) **+1.3%, +4.7%, and +0.4%** over previous SOTA NAS baselines

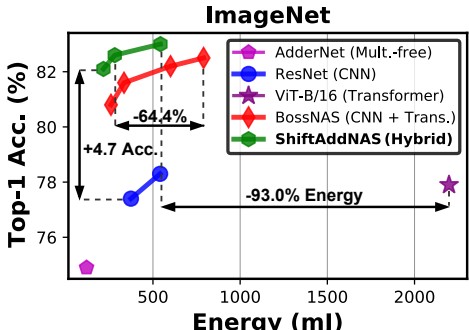

Figure 6: Accuracy vs. energy costs of ShiftAddNAS over baselines.

BossNAS, VITAS, and Autoformer, respectively, under a comparable or even less MACs. Moreover, considering looser MACs constraints, we follow BossNAS to remove the downsampling in the last stage, resulting in ShiftAddNAS-T1 with a accuracy of **82.7%** and 6.4G MACs that surpasses T2T-ViT and BoTNet-S1-59 by **+0.8%** and **+1.0%** at even less MACs. By directly testing on larger input resolutions without finetuning, ShiftAddNAS-T1↑ (w/ $256^2$ input resolution) offers an accuracy of **83.0%**, surpassing BossNAS-T1 and Autoformer-B by **+0.8%** and **+0.6%** with comparable

Table 4: Comparison with SOTA baselines on ImageNet classification task.

| Model | Top-1 Acc. | Top-5 Acc. | Params | Res. | MACs | #Mult. | #Add | #Shift | Model Type |
|---|---|---|---|---|---|---|---|---|---|
| BNN | 55.8% | 78.4% | 26M | $224^2$ | 3.9G | 0.1G | 3.9G | 3.8G | Mult.-free |
| AdderNet | 74.9% | 91.7% | 26M | $224^2$ | 3.9G | 0.1G | 7.6G | 0 | Mult.-free |
| AdderNet-PKKD | 76.8% | 93.3% | 26M | $224^2$ | 3.9G | 0.1G | 7.6G | 0 | Mult.-free |
| DeepShift-Q | 70.7% | 90.2% | 26M | $224^2$ | 3.9G | 0.1G | 3.9G | 3.8G | Mult.-free |
| DeepShift-PS | 71.9% | 90.2% | 52M | $224^2$ | 3.9G | 0.1G | 3.9G | 3.8G | Mult.-free |
| ShiftAddNet | 72.3% | - | 64M | $224^2$ | 10G | 0.1G | 16G | 3.9G | Mult.-free |
| ResNet-50 | 76.1% | 92.9% | 26M | $224^2$ | 3.9G | 3.9G | 3.9G | 0 | CNN |
| ResNet-101 | 77.4% | 94.2% | 45M | $224^2$ | 7.6G | 7.6G | 7.6G | 0 | CNN |
| SENet-50 | 79.4% | 94.6% | 26M | $224^2$ | 3.9G | 3.9G | 3.9G | 0 | CNN |
| SENet-101 | 81.4% | 95.7% | 45M | $224^2$ | 7.6G | 7.6G | 7.6G | 0 | CNN |
| ViT-B/16 | 77.9% | - | 86M | $384^2$ | 18G | 18G | 17G | 0 | Transformer |
| ViT-L/16 | 76.5% | - | 304M | $384^2$ | 64G | 64G | 63G | 0 | Transformer |
| DeiT-T | 74.5% | - | 6M | $224^2$ | 1.3G | 1.3G | 1.3G | 0 | Transformer |
| DeiT-S | 81.2% | - | 22M | $224^2$ | 4.6G | 4.6G | 4.6G | 0 | Transformer |
| VITAS | 77.4% | 93.8% | 13M | $224^2$ | 2.7G | 2.7G | 2.7G | 0 | Transformer |
| Autoformer-S | 81.7% | 95.7% | 23M | $224^2$ | 5.1G | 5.1G | 5.1G | 0 | Transformer |
| BoT-50 | 78.3% | 94.2% | 21M | $224^2$ | 4.0G | 4.0G | 4.0G | 0 | CNN + Trans. |
| BoT-50 + SE | 79.6% | 94.6% | 21M | $224^2$ | 4.0G | 4.0G | 4.0G | 0 | CNN + Trans. |
| HR-NAS | 77.3% | - | 6.4M | $224^2$ | 0.4G | 0.4G | 0.4G | 0 | CNN + Trans. |
| BossNAS-T0 | 80.5% | 95.0% | 38M | $224^2$ | 3.5G | 3.5G | 3.5G | 0 | CNN + Trans. |
| BossNAS-T0 + SE | 80.8% | 95.2% | 38M | $224^2$ | 3.5G | 3.5G | 3.5G | 0 | CNN + Trans. |
| **ShiftAddNAS-T0** | **82.1%** | **95.8%** | 30M | $224^2$ | 3.7G | 2.7G | 3.8G | 1.0G | Hybrid |
| **ShiftAddNAS-T0↑** | **82.6%** | **96.2%** | 30M | $256^2$ | 4.9G | 3.6G | 4.9G | 1.4G | Hybrid |
| T2T-ViT-19 | 81.9% | - | 39M | $224^2$ | 8.9G | 8.9G | 8.9G | 0 | Transformer |
| TNT-S | 81.3% | 95.6% | 24M | $224^2$ | 5.2G | 5.2G | 5.2G | 0 | Transformer |
| Autoformer-B | 82.4% | 95.7% | 54M | $224^2$ | 11G | 11G | 11G | 0 | Transformer |
| BoTNet-S1-59 | 81.7% | 95.8% | 28M | $224^2$ | 7.3G | 7.3G | 7.3G | 0 | CNN + Trans. |
| BossNAS-T1 | 82.2% | 95.8% | 38M | $224^2$ | 8.0G | 8.0G | 8.0G | 0 | CNN + Trans. |
| **ShiftAddNAS-T1** | **82.7%** | **96.1%** | 30M | $224^2$ | 6.4G | 5.4G | 6.4G | 1.0G | Hybrid |
| **ShiftAddNAS-T1↑** | **83.0%** | **96.4%** | 30M | $256^2$ | 8.5G | 7.1G | 8.5G | 1.4G | Hybrid |

Table 5: Ablation study of ShiftAddNAS w/ (1) naive and (2) heterogeneous weight sharing.

| ShiftAddNAS | Kendall $\tau$ | Pearson $R$ | Spearman $\rho$ | Top-1 Acc. | Params | Energy | Latency |
|---|---|---|---|---|---|---|---|
| w/ Naive WS | 0.49 | 0.67 | 0.69 | 81.3% | 30M | 440mJ | 387ms |
| **w/ HWS** | **0.54** | **0.75** | **0.74** | **82.7%** | **30M** | **413mJ** | **252ms** |

or even less MACs, respectively. Finally, we compare ShiftAddNAS with representative baselines of various model types in terms of accuracy and energy cost in Fig. 6, where each line represents a series searched/designed models with various FLOPs constraints. We can see that ShiftAddNAS consistently outperforms all the baselines, on average offering a **+0.8%** ∼ **+7.7%** higher accuracy and **24%** ∼ **93%** energy savings. Specifically, our ShiftAddNAS on average achieves a **+0.8%** higher accuracy and **30%** energy savings against the competitive NAS baseline BossNAS.

## 4.4 ABLATION STUDIES OF SHIFTADDNAS

We conduct ablation studies on ShiftAddNAS's heterogeneous weight sharing (HWS) strategy, as shown in Tab. 5. *First*, for searching on ImageNet, we use three ranking correlation metrics: Kendall $\tau$, Pearson $R$, and Spearman $\rho$, to measure the ranking correlation between ShiftAddNAS w/ and w/o HWS and find that the former leads to a higher ranking correlation than the naive WS. *Second*, the proposed HWS leads to more accurate searched subnets. Specifically, the searched subnet achieves a **+1.4%** higher accuracy than that of naive WS, at comparable or even smaller energy and latency costs. Also, HWS effectively reduces the supernet size from 615M (w/o WS) to 364M (41% savings). This set of ablation studies validate the effectiveness of our proposed HWS strategy.

## 5 CONCLUSION

We propose ShiftAddNAS for searching for multiplication-reduced NNs incorporating both powerful yet costly multiplications and efficient yet less powerful shift and add operators for marrying the best of both worlds. ShiftAddNAS is made possible by integrating: (1) the first hybrid search space that incorporates both multiplication-based and multiplication-free operators for searching for more accurate and efficient NNs; and (2) a novel heterogeneous weight sharing strategy that allows different operators to follow heterogeneous distributions for alleviating the dilemma of either inefficient search or inconsistent architecture ranking when applying NAS for hybrid NNs. Extensive experiments on both NLP and CV tasks demonstrate the superior accuracy and energy efficiency of ShiftAddNAS's searched NNs over various SOTA baselines. Our ShiftAddNAS has opened up a new perspective in searching for more accuracy and efficient NNs.

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

## A    EVALUATE THE SEARCH COST

We further supply the total search cost of ShiftAddNAS on both NLP tasks and CV tasks to Table 6 and 7, respectively. For NLP tasks, with one Nvidia V100 GPU, ShiftAddNAS uses on average 9.3 GPU days (Gds) for searching which is comparable to HAT (Wang et al., 2020a) and 9,821× less than the Evolved Transformer (So et al., 2019); For CV tasks, ShiftAddNAS uses on average 8.9 Gds for searching which is 11% and 82% less than DARTS (Liu et al., 2018) and BossNAS (Li et al., 2021), respectively. In addition, we provide a concrete breakdown analysis of ShiftAddNAS search cost in Table 8. For NLP tasks, ShiftAddNAS uses on average 8.5 Gds for supernet training and 0.8 Gds for architecture searching; For CV tasks, ShiftAddNAS uses on average 7.7 Gds for supernet training and 1.2 Gds for architecture searching.

Table 6: Search cost on NLP tasks.

| Methods | Search Cost |
|---|---|
| Evolved Trans. | 91,334 Gds |
| HAT | 9.3 Gds |
| **ShiftAddNAS** | 9.3 Gds |

Table 7: Search cost on CV tasks.

| Methods | Search Cost |
|---|---|
| DARTS | 50 Gds |
| BossNAS | 10 Gds |
| **ShiftAddNAS** | 8.9 Gds |

Table 8: Breakdown analysis of the search cost of ShiftAddNAS.

| | NLP | CV |
|---|---|---|
| Supernet Train | 8.5 Gds | 7.7 Gds |
| Arch. Search | 0.8 Gds | 1.2 Gds |

## B    VISUALIZATION OF THE HETEROGENEOUS WEIGHT DISTRIBUTIONS

For better understanding of the proposed heterogeneous weight sharing strategy, we further supply the visualization of the heterogeneous weight distributions in **Conv/Add/Shift** layers, respectively, as shown in the Fig. 7.

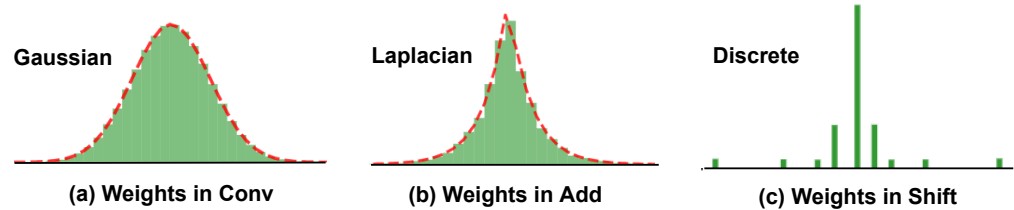

(a) Weights in Conv    (b) Weights in Add    (c) Weights in Shift

Figure 7: Visualization of the heterogeneous weight distributions in **Conv/Add/Shift** layers.

## C    VISUALIZATION OF THE SEARCHED ARCHITECTURE

For better understanding of the searched architecture, we supply the visualization of the searched architecture by ShiftAddNAS in Fig. 8. The searched architecture contains two adder blocks and one shift block, and achieves 82.8% top-1 test accuracy on ImageNet with 8.4G MACs (Mult: 7.4G; Add: 9.1G; Shift: 0.5G). Also, the searched architecture prefers **Conv** as early blocks while consider **Attn** as later blocks, which is also consistent with the previous empirical observation that early convolutions help the overall performance (Xiao et al., 2021).

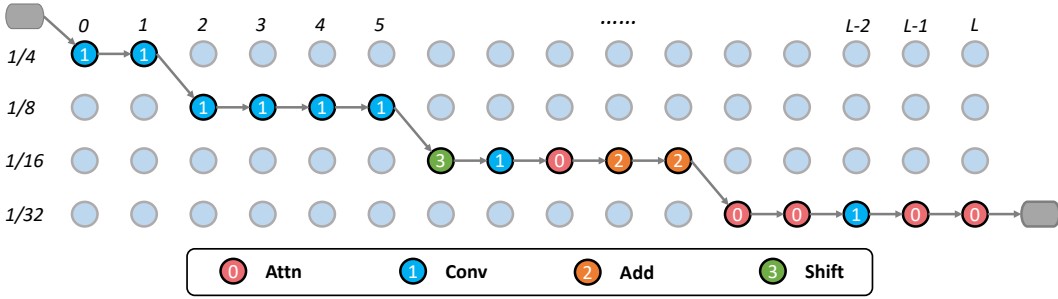

Figure 8: Visualization of the searched architecture with 82.8% top-1 test accuracy on ImageNet.

