# OpenReview forum: "ShiftAddNAS: Hardware-Inspired Search for More Accurate and Efficient Neural Networks"
_ICLR.cc/2022/Conference — ICLR 2022 Submitted_

### Official Review · Reviewer_1Due · 2021-11-02

**Correctness:** 3
**Technical Novelty And Significance:** 2
**Empirical Novelty And Significance:** 2
**Recommendation:** 5
**Confidence:** 4

**Main Review:**

Strengths:
1. I think it is quite interesting to integrate hardware-efficient shift/add operations into the search space of NAS, which can potentially improve the efficiency of neural networks on hardware.
2. The motivation of heterogeneous weight sharing is clear and reasonable.

Weaknesses:
1. Insufficient technical contributions. The idea of heterogeneous weight sharing is not new. As far as I know, when training super-nets with flexible quantization bits (e.g., [1]), a similar strategy has already been used: maintain a shared weight pool, fp32 directly uses the weight, int8/4/2 use the quantized weights. Compared to the previous strategy, this work has additional regularization terms (Gaussian for Conv ops and Laplacian for Add ops). To convince me that this modification provides sufficient technical contributions, I would expect to see solid experimental studies showing clear and significant improvements from these regularization terms. In addition, as the search space only has three different types of operations (according to Figure 4), maintaining the weight separately for different types of blocks and having three activate paths seems to be doable. I would expect to see comparisons with this approach as well.
2. Weak results. With a more hardware-efficient design space and the proposed heterogeneous weight sharing, I expect to see clear and significant improvements compared to previous SOTA methods. However, the improvements seem to be marginal. For example, in Table 3, ShiftAddNAS achieves the same BLEU score on WMT'14 En-De as HAT while providing only 1.05x latency reduction. In Table 4, the improvements of ShiftAddNAS also look marginal compared to strong baselines (e.g., EfficientNet, FBNetV3, etc). In addition, it would be helpful to provide the results of ShiftAddNAS with smaller MACs in Table 4 (e.g., <600M, <300M, etc).

[1] Bai, Haoping, et al. "BatchQuant: Quantized-for-all Architecture Search with Robust Quantizer." arXiv preprint arXiv:2105.08952 (2021).

**Summary Of The Paper:**

This paper introduces ShiftAddNAS that performs neural architecture search on a hybrid design space that contains conv operations, attention operations, and hardware-efficient shift/add operations. A heterogeneous weight sharing strategy is proposed to share weights between different types of blocks.

**Summary Of The Review:**

Overall, I think this paper explores an interesting topic and potentially can benefit real-world efficient deep learning applications. However, I find the technical contributions and empirical results presented in this version are not strong enough. Thus, I recommend rejection.

---

> ### Author Response · Authors · 2021-11-20
> **Response to Reviewer 1Due**
>
> Thank you for your review. Below we have provided the detailed responses for clarification.
>
> ---
> **W1: Insufficient technical contributions? Comparison with BatchQuant (NeurIPS’21)? Benefits of heterogeneous weight sharing? Search space only has three different types? Are activating all paths doable?**
>
> * As other reviewers have recognized, we humbly argue that we are the first to develop a dedicated framework to enable automatically searching for efficient hybrid NNs by integrating (1) a hybrid hardware-inspired search space that incorporates both multiplication-based operators and multiplication-free operators, and (2) a heterogeneous weight sharing (WS) strategy that allows different operators to follow different weight distributions, alleviating the dilemma between inefficient search and inconsistent architecture ranking.
> * We humbly clarify that our heterogeneous WS for different weight distributions is orthogonal to and different from the WS of different quantization bits as presented in the mentioned BatchQuant (NeurIPS’21). Specifically, our work can be applied on top of BatchQuant, and while ours is to enable weight sharing supernet training via a learnable transformation, BatchQuant is for DNN quantization and targets weight sharing for the same parameters under different bitwidths. In addition, as a soft reminder, NeurIPS’21 released its camera ready papers ***after Oct. 26 and will be held on Dec. 6***. According to the ICLR’22 policy that *"We consider papers contemporaneous if they are published (available in online proceedings) within the last four months. That means, if a paper was published at a peer-reviewed venue on or after June 5, 2021, it can be considered as a concurrent work"*, i.e., our work is contemporaneous to BatchQuant.
> * We humbly clarify that we do show the ablation studies of the heterogeneous WS in Sec. 4.4. The proposed heterogeneous WS effectively reduces the supernet size from 615M (w/o WS) to 364M (41% savings), and also leads to a higher ranking correlation and more accurate searched subnets (+1.4% higher accuracy) than the naive WS.
> * It is worth noting that our supernet has four types of operators (Attn, Conv, Shift, and Add) and they are searchable in all four resolution branches as clearly shown in Fig. 2 and Table 2; Thus, activating all the paths will lead to a huge supernet with 615M parameters and 416GB GPU memory requirement when using 256 batch size, which cannot be fed into a server with eight Nvidia 32GB V100 GPUs and can be practically challenging for many research groups. Furthermore, looking forward, there would be more hardware friendly operators invented, making our proposed framework even more useful.
>
> ---
> **W2: Weak results and marginal improvement? Comparison to strong baselines (e.g., FBNet) with smaller MACs (e.g., <600M, <300M, etc)?**
>
> We humbly disagree with your comments of weak results. For NLP tasks, ShiftAddNAS with a full precision achieves 11.8% ~ 73.6% FLOPs reductions and up to 69.1% and 69.2% energy and latency savings while offering a comparable or even better BLEU score (-0.3 ~ +1.1), over all the full precision baselines (Transformer, Evolved Transformer, and HAT). Furthermore, ShiftAddNAS (8-bit) more aggressively achieves +1.8 ~ +4.9 BLEU scores improvements over the Lite Transformer (8-bit), while offering up to 82.7% FLOPs reductions. For CV tasks, ShiftAddNAS consistently outperforms all the baselines, on average offering a +0.8% ~ +7.7% higher accuracy and 24% ~ 93% energy savings.
>
> To better address your comment and fairly compare with the mentioned baselines with smaller MACs, we implement our proposed hybrid search space and heterogeneous WS on top of the FBNet search space, and evaluate the performance on CIFAR-10/100 instead of ImageNet due to the limited rebuttal timeslots. As shown in the table below, our ShiftAddNAS consistently boosts a +0.74% and +0.58% accuracy over FBNet and leads to 33.8% and 38.6% latency savings on CIFAR-10 and CIFAR-100, respectively.
>
> | Datasets | Methods | Accuracy | MACs | #Mult. | #Add | #Shift |  Latency Saving |
> |---|---|---|:---:|:---:|:---:|:---:|:---:|
> | CIFAR-10 | FBNet | 95.09% | 47M | 47M | 47M | 0 | - |
> | CIFAR-10 | ShiftAddNAS | 95.83% (+0.74%) | 47M | 17M | 58M | 19M |  33.8% |
> | CIFAR-100 | FBNet | 77.86% | 55M | 55M | 55M | 0 | - |
> | CIFAR-100 | ShiftAddNAS | 78.64% (+0.58%) | 52M | 22M | 62M | 21M | 38.6% |
>
> Furthermore, in the efficient neural network architecture community, it is recognized there exists a performance gap between AdderNet/ShiftAddNet and CNN on ImageNet. Our proposed ShiftAddNAS successfully closes the aforementioned gap or even produces hybrid models that surpass the SOTA CNN, Transformer, or CNN-Transformer designs in terms of both accuracy and efficiency, contributing new insights to the community.

---

> > ### Comment · Reviewer_1Due · 2021-11-27
> > **Thanks for the response**
> >
> > I appreciate the authors' response. I still have the following concerns that the authors do not address in their rebuttal:
> >
> > 1. If maintaining separate weights for different ops and having 4 active paths, it seems that the GPU memory is only ~4x larger. It is not clear to me why it requires 416GB GPU memory when using 256 batch size. I think it is important to compare the proposed method with this simple approach. If the performance degrades, it may not be worthwhile to sacrifice the performance to get this GPU memory saving. There are many simple ways to save the GPU memory such as fp16 training, sub-linear memory training [1], etc.
> >
> > 2. The proposed method is a bit similar to one-shot NAS with quantization (e.g., shift/add ops -> quantized ops). I suggest the authors add comparisons with these methods.  I understand the authors may not have sufficient time to get the results before the end of the discussion period. It is just a suggestion.
> >
> > [1] Chen, Tianqi, et al. "Training deep nets with sublinear memory cost." arXiv preprint arXiv:1604.06174 (2016).

---

> > > ### Author Response · Authors · 2021-11-28
> > > **Response to Reviewer 1Due (Part 1)**
> > >
> > > We thank the reviewer for your feedback and additional comments especially during this holiday time. Below are our detailed responses:
> > >
> > > ---
> > > **Q1 - 1: GPU memory is only ~4x larger when activating 4 paths, why require 416G GPU memory?**
> > >
> > > Apart from the four paths (Attn, Conv, Shift, and Add), our supernet for CV tasks also contains four resolution branches, as shown in Fig. 2 / Table 2 of our submitted manuscript and as explained in the first run of our response to W1, leading to a 16x larger memory requirement as compared to searched architectures if not considering channel reduction as elaborated below:
> > > When training the supernet of 615M parameters w/o weight sharing, we obtained the 416G GPU memory cost via measuring on 16 Nvidia V100 GPUs, which are usually hard to get and will cost $4,000 (a total of 1000 GPU hours) for searching and training the ShiftAddNAS-T0/T1 as in Table 4 of our submitted, based on the price of Amazon EC2 P3 instances.  Therefore, we propose to adopt weight sharing based training with 8 GPUs, which halves the training cost. As a reference, when training the supernet of 364M parameters under heterogeneous weight sharing, it requires a total of (30.75G*8=)246G GPU memory (30.75G for each of the eight GPUs).
> > >
> > > Finally, we humbly argue that our heterogeneous weight sharing methodology would be even more useful as more efficient and heterogeneous operators are developed, leading to reduced memory requirement and better searched networks. We believe the reviewer agrees that it is not practical to always seek solutions by merely scaling up the computational resources, motivating more efficient techniques like our proposed one and other prior arts.
> > >
> > > ---
> > > **Q1 - 2: Comparison with ShiftAddNAS without weight sharing?**
> > >
> > > Following your suggestions of adopting approaches to activate all paths, we tried ShiftAddNAS without weight sharing (WS) by training supernet with reduced channel dimensions and FP16 optimization in order to fit into a server with eight Nvidia 32GB V100 GPUs. Specifically, we evaluated the searched network on ImageNet and summarized the required performance comparison in the table below, from which we see that ShiftAddNAS with heterogeneous weight sharing (ShiftAddNAS w/ HWS; i.e., the proposed one) achieves up to +1.2% higher top-1 accuracy (82.7% vs. 81.5%) than ShiftAddNAS without weight sharing (ShiftAddNAS w/o WS), under comparable MACs (6.4G vs. 6.4G) and latency cost (252ms vs. 269ms) when measuring on the Eyeriss accelerator. This set of experiments validate the effectiveness of the proposed heterogeneous weight sharing strategy.
> > >
> > > | Methods | Top-1 Acc. | MACs | #Mult. | #Add | #Shift |  Latency | Supernet Size |
> > > |---|---|:---:|:---:|:---:|:---:|:---:|---|
> > > | ShiftAddNAS w/o WS | 81.5% | 6.4G | 5.6G | 6.8G | 0.4G | 269ms | 615M |
> > > | ShiftAddNAS w/ HWS | 82.7% (+1.2%) | 6.4G | 5.4G | 6.4G | 1.0G | 252ms | 364M (41% savings)

---

> > > > ### Comment · Reviewer_1Due · 2021-11-30
> > > > **Score updated**
> > > >
> > > > I appreciate the authors' clarifications and additional results, which partially address my concerns. I have increased the score to reflect this. However, I am still confused why the authors used the multi-resolution supernet design instead of the commonly used single-branch supernet design. I think it is not necessary/beneficial for NAS to search in such a large and complicated search space ([1]). In the future, when we have more types of operations, I think it is important to identify a compact search space instead of directly searching in the large design space. I am worried that if the proposed method is only effective when applied to very complicated search spaces, as suggested by current results, the practical value of the proposed method might be very limited.
> > > >
> > > > [1] Radosavovic, Ilija, et al. "Designing network design spaces." Proceedings of the IEEE/CVF Conference on Computer Vision and Pattern Recognition. 2020.

---

> > > > > ### Author Response · Authors · 2021-11-30
> > > > > **Response to Reviewer 1Due**
> > > > >
> > > > > We are glad that our rebuttal has addressed most of your concerns and appreciate your updated score. Below are our detailed clarifications for your confusion points:
> > > > >
> > > > > ---
> > > > > **C1: Why did the authors use the multi-resolution supernet design instead of the commonly used single-branch supernet design?**
> > > > >
> > > > > We adopt the multi-resolution supernet design because it’s promising to generate decent network architectures for most common CV tasks including classification, object detection, and semantic segmentation according to the latest SOTA NAS works [1][2], which validate that combining the information from different spatial resolutions are essential for CV tasks for better performance. For example, in FBNetV5 the authors pointed out “Surprisingly, the CLS (short for image classification) model contains a lot of blocks from higher resolutions. This contrasts the mainstream models that only stack layers sequentially” and “our searched CLS model demonstrates stronger performance than sequential architectures, this may open up a new direction for the classification model design”, when discussing their visualized network architectures as shown in Fig. 3 of their manuscript.
> > > > >
> > > > > In addition, our proposed method can generally serve as a play-and-plug module to be applied on top of different types of supernets, and in this paper we demonstrate its applicability to both the single-branch supernet design for NLP tasks and the multi-resolution supernet design for CV tasks, as shown in Fig. 2 of our submitted manuscript. We also consider applying our method to more supernets and tasks as our promising future work.
> > > > >
> > > > > Moreover, from a technical point of view, the adoption of multi-resolution supernet for CV tasks helps to alleviate the mismatch between convolutions and attention blocks on the spatial resolution, as analyzed in [3]. Specifically, convolution-based CNNs process images in stages with various spatial resolutions, while attention-based Transformers do not change the image patches and remain the same resolution for all layers. Such a mismatch necessitates both a downsampling knob and multi-resolution supernet for effectively searching for hybrid network architectures, as demonstrated in the recent supernet designs [1][3].
> > > > >
> > > > > [1] Ding et al., HR-NAS: Searching Efficient High-Resolution Neural Architectures with Lightweight Transformers, CVPR’21
> > > > >
> > > > > [2] Wu et al., FBNetV5: Neural Architecture Search for Multiple Tasks in One Run, arXiv’21
> > > > >
> > > > > [3] Li et al., BossNAS: Exploring Hybrid CNN-transformers with Block-wisely Self-supervised Neural Architecture Search, ICCV’21
> > > > >
> > > > > ---
> > > > > **C2: Whether the proposed method is only effective when applied to very complicated search spaces?**
> > > > >
> > > > > Following your previous suggestions, we have also implemented our proposed hybrid search space and heterogeneous weight sharing on top of the FBNet search space which has the single-branch supernet design, and evaluated the performance on CIFAR-10/100. As shown in the table below, our ShiftAddNAS consistently boosts a +0.74% and +0.58% accuracy over FBNet and leads to 33.8% and 38.6% latency savings on CIFAR-10 and CIFAR-100, respectively. This set of experiments also validate that the proposed methods can also generalize to lightweight supernet designs without multi-resolution branches.
> > > > >
> > > > > | Datasets | Methods | Accuracy | MACs | #Mult. | #Add | #Shift |  Latency Saving |
> > > > > |---|---|---|:---:|:---:|:---:|:---:|:---:|
> > > > > | CIFAR-10 | FBNet | 95.09% | 47M | 47M | 47M | 0 | - |
> > > > > | CIFAR-10 | ShiftAddNAS | 95.83% (+0.74%) | 47M | 17M | 58M | 19M |  33.8% |
> > > > > | CIFAR-100 | FBNet | 77.86% | 55M | 55M | 55M | 0 | - |
> > > > > | CIFAR-100 | ShiftAddNAS | 78.64% (+0.58%) | 52M | 22M | 62M | 21M | 38.6% |

---

> > > ### Author Response · Authors · 2021-11-28
> > > **Response to Reviewer 1Due (Part 2)**
> > >
> > > ---
> > > **Q2: Comparison with one-shot NAS with quantization?**
> > >
> > > To the best of our knowledge, we find two related works regarding one-shot NAS with mixed quantization: (1) BatchQuant [1] and (2) APQ [2]. Both these two works only report results on ImageNet. Although we achieve a much better accuracy (ShiftAddNAS: 82.1% (see Table 4 of our submitted manuscript) vs. BatchQuant: 75% vs, APQ: 75.1%), the FLOPs ranges of their searched networks are smaller than ours since they consider a MobileNet-like search space as the backbone (ShiftAddNAS: 3.7G vs. BatchQuant: 200M vs. APQ: 400M). As such, we cannot directly compare them.
> > >
> > > To better address your comments and fairly compare with NAS with quantization related methods, we consider a benchmark with Auto-NBA [3] which achieves better results than APQ [2] and also automatically searches for networks with various quantization bitwidths via differentiable NAS and weight sharing strategy. We have evaluated the performance on CIFAR-10/100 instead of ImageNet, due to the limited given time (i.e., from Nov 26 to Nov 29, as we provided our response to your first-run review on Nov 20, and received your new request for this comparison on Nov 26, while our discussion needs to finish before Nov 29) and by confirming with Auto-NBA’s authors on their MAC information. As shown in the table below, our ShiftAddNAS consistently boosts a +0.34% and +1.21% accuracy over Auto-NBA on CIFAR-10 and CIFAR-100, respectively, under comparable or even lower MACs. This set of experiments validate that our proposed ShiftAddNAS leads to searched networks with a better accuracy-efficiency tradeoff than SOTA NAS with quantization methods, i.e., Ours > Auto-NBA [3] > APQ [2].
> > >
> > > | Datasets | Methods | Accuracy | MACs | #Mult. | #Add | #Shift |
> > > |---|---|---|:---:|:---:|:---:|:---:|
> > > | CIFAR-10 | Auto-NBA | 95.49% | 51M | 51M | 51M | 0 |
> > > | CIFAR-10 | ShiftAddNAS | 95.83% (+0.34%) | 47M | 17M | 58M | 19M |
> > > | CIFAR-100 | Auto-NBA | 77.43% | 55M | 55M | 55M | 0 |
> > > | CIFAR-100 | ShiftAddNAS | 78.64% (+1.21%) | 52M | 22M | 62M | 21M |
> > >
> > > Finally, we will review and properly cite/comment and compare with your mentioned BatchQuant [1] and APQ [2] works in our final revision, and would appreciate your feedback on our above response and clarification.
> > >
> > > ---
> > > [1] Bai et al., BatchQuant: Quantized-for-all Architecture Search with Robust Quantizer, NeurIPS’21
> > >
> > > [2] Wang et al., APQ: Joint Search for Network Architecture, Pruning and Quantization Policy, CVPR’20
> > >
> > > [3] Fu et al., Auto-NBA: Efficient and Effective Search Over the Joint Space of Networks, Bitwidths, and Accelerators, ICML’21

---

> ### Author Response · Authors · 2021-11-26
> **We Sincerely Look Forward to Your Post Rebuttal Feedback!**
>
> Dear Reviewer 1Due,
>
> We are following up to check whether our rebuttal responses have addressed your comments/concerns, and would be appreciative if you could let us know your feedback, thanks and happy holiday!
>
> Best Regards,
> Authors of Paper #2246

---

### Official Review · Reviewer_evhx · 2021-11-02

**Correctness:** 3
**Technical Novelty And Significance:** 3
**Empirical Novelty And Significance:** 3
**Recommendation:** 6
**Confidence:** 5

**Main Review:**

Pros:
1. This paper is innovative. The search space contains both multiplication-free and multiplication-based operators.
2. The results are the state-of-the-arts on both NLP and CV tasks.

Cons:
1. What is the meaning of Naïve WS in Table 5? It means weight sharing between operations without transformation or no weight sharing between operations? Generally, we only share weights of the same operator between different architectures, why sharing weights between operators here? In most cases, the more weights are shared, the lower the Kendall-tau is.
2. A learnable transformation kernel is used to translate the convolution to shift or adder operation. What if we just apply this kernel to the whole convolution neural network? Can we get an energy efficient adder or shift neural network without much accuracy loss with this kernel? There should be more analyses and experiments to show the effectiveness of this kernel.
3. In Table 4, the number of Add in ShiftAddNAS is always the same as MACs. In my opinion, this means no adder operator is used, since one MAC actually means two Add for adder operation (just like the AdderNet in Table 4). Why there is no adder operator at all? There should be figures to show your final searched models.


**Summary Of The Paper:**

This paper propose a hybrid search space including both multiplication-based and multiplication-free operators. The authors also give a novel weight sharing strategy to enable effective weight sharing among different operators.

**Summary Of The Review:**

The idea of this paper is novel and the results are satisfying, but there are more details should be provided.

---

> ### Author Response · Authors · 2021-11-20
> **Response to Reviewer evhx**
>
> We greatly appreciate your careful review and constructive suggestions, and our detailed responses are as follows:
>
> ---
> **W1: What does naive WS mean in Table 5? Why share weights among operators instead of architectures? The more weights are shared, the lower Kendall-tau is?**
>
> * The naive weight sharing (WS) means weight sharing between operations without any transformation so that different operators follow the same weight distribution, i.e., a typical weight sharing method.
> * In one-shot NAS, different subnet candidates can directly inherit their weights from the supernet by activating different operators; here we share weights of various operators so that the weights of different subnets can also be shared to enable favorable supernet training costs.
> * Yes, we agree that the more weights are shared, the lower the Kendall-tau is, due to the increased difficulty of differentiating various subnets. On top of this general trend, our proposed heterogeneous WS allows different operators to follow different weight distributions  (i.e., their uniquely favorable distribution) for reducing the difficulty of differentiating various subnets within the supernet, thus leading to a higher Kendall-tau correlation than naive WS.
>
> ---
> **W2: Can we directly apply the learnable transformation kernel to the whole convolutional neural network to get an energy-efficient adder or shift neural network without much accuracy loss?**
>
> Great question! The transformation kernel cannot be directly applied to all pretrained CNNs because it contains learnable parameters and thus should be jointly trained with the supernet. That said, if we jointly optimize the supernet and transformation kernel, then we can directly apply this kernel to supernet for getting a purely ShiftNet or AdderNet which however leads to about 3% and 4% accuracy drop, respectively. This set of experiments further validate the effectiveness of the proposed transformation kernel, and the necessity of a hybrid search space.
>
> ---
> **W3: In Table 4, the number of Add in ShiftAddNAS is always the same as MACs. Why is there no adder operator?**
>
> We appreciate the reviewer for his/her deep understanding of our work. In table 4, we report the rank #1 searched architecture with the highest accuracy. Meanwhile, our rank #2 searched architecture contains two adder blocks and achieves a 82.8% top-1 accuracy on ImageNet with 8.4G MACs (#Mult: 7.4G; #Add: 9.1G; #Shift: 0.5G), and we have also supplied the visualization of this architecture to the Appendix C of the revised manuscript.

---

> > ### Comment · Reviewer_evhx · 2021-11-25
> > **Post Rebuttal Update**
> >
> > I appreciate the feedback from the authors. The rebuttal properly address most of my concerns and I tend to keep my original ratings. A few minor comments:
> > - More visualizations for the discovered architectures listed in Table 4 are highly encouraged.
> > - Please provide more discussions for the searched architectures. For example, why rank#1 searched models contain no adder operator? Are there any insights from the searched models about the ratio of different operators?

---

> > > ### Author Response · Authors · 2021-11-25
> > > **Response to Reviewer evhx**
> > >
> > > We thank the reviewer for the timely feedback and additional comments/suggestions! It is our pleasure to address your concerns and get your additional comments to help us further improve the paper quality. Below are our detailed responses:
> > >
> > > ---
> > > **C1: More visualizations for the discovered architectures listed in Table 4 are highly encouraged.**
> > >
> > > We appreciate your comments and also agree on the importance of more visualizations. Since the manuscript cannot be updated now, we list the architecture shown in Table 4 below and will also supply more visualization to the final revision.
> > >
> > > Conv → Conv → Conv → Shift → Conv → Conv → Shift → Conv → Attn → Shift → Shift → Attn → Attn → Conv → Attn → Attn
> > >
> > > ---
> > > **C2: More discussions for the searched architectures, e.g., why rank #1 searched models contain no adder operator? Are there any insights from the searched models about the ratio of different operators?**
> > >
> > > Great question! Actually, the performance of the searched top architectures are quite close (e.g., the accuracy difference between rank #1 and rank #2 is smaller than 0.2%), and we also quantitatively measure the ratio breakdown of different operators in our searched top-10 architectures (Attn: 30%; Conv: 43%; Shift: 15%; Add: 12%). We see that the overall ratio of Add operators is quite comparable with that of Shift operators, thus our understanding is that the searched top architectures benefit from a relatively more balanced combination/contribution of all operators to push forward the frontier of accuracy-efficiency tradeoff. We will add more discussion regarding this aspect in the final revision. Furthermore, your constructive suggestion of showing such a ratio breakdown in searched top architectures further validates the necessity of adopting a hybrid search space.
> > >
> > > Finally, another insight from the general trends of the operator ratio breakdown is that, as the ranking of searched architectures drops, the ratio of Shift and Add operators will increase, which is consistent with our previous reply to W2, i.e., a purely ShiftNet or AdderNet will inevitably lead to accuracy drop. The above analysis also helps to validate the correctness of our search algorithm.
> > >
> > > In a summary, we greatly appreciate your comments which have inspired us to better clarify our work’s contributions/insights/experiments and validate our proposed methods. Happy Thanksgiving!

---

> > > > ### Comment · Reviewer_evhx · 2021-11-30
> > > > **Thanks for the update**
> > > >
> > > > Thanks for the response and I think this is an interesting paper that can be accepted.

---

> > > > > ### Author Response · Authors · 2021-11-30
> > > > > **Response to Reviewer evhx**
> > > > >
> > > > > Thank you for your appreciation of our work! We will include your suggested visualization and experiments in our final version to further improve the clarity and strengthen our contributions.

---

### Official Review · Reviewer_KyCd · 2021-11-02

**Correctness:** 4
**Technical Novelty And Significance:** 3
**Empirical Novelty And Significance:** 3
**Recommendation:** 6
**Confidence:** 4

**Main Review:**

Strengths:
+ The specially designed search space that incorporates both multiplication-based and multiplication-free operators is new.
+ The analysis on distributions of convolution and add operators is reasonable.
+ The proposed weight sharing strategy can enable effective weight sharing among different operators that follow heterogeneous distributions.

Weaknesses:
- The weight sharing strategy only considers convolution and add operations. What's the distribution of Shift operation? Should shift op be considered together?
- In the experiment, lacking comparison with ShiftAddNet[1].

[1] ShiftAddNet: A Hardware-Inspired Deep Network. NeurIPS 2020.

**Summary Of The Paper:**

This paper proposes a NAS method for both multiplication-based and adder-based networks. The contributions mainly lie on hybrid search space and new weight sharing strategy. The experimental results shows the method can obtain energy-efficient networks with high performance.

**Summary Of The Review:**

The main contribution of the paper is a NAS framework for multiplication-based and multiplication-free operators. The weaknesses do not affect the main contribution. I recommend accptance for it.

---

> ### Author Response · Authors · 2021-11-20
> **Response to Reviewer KyCd**
>
> We greatly appreciate your careful review and constructive suggestions, and have provided our detailed responses below:
>
> ---
> **W1: The weight sharing strategy should also consider shift operations? What is the distribution of shift weights?**
>
> Yes, we do consider weight sharing for the shift operations, whose weights are derived from directly quantizing the shared weights to powers of two as illustrated in Fig. 4 (a). In particular, the weight distribution of the shift operations is a discrete one, and we have provided its detailed visualization to Fig. 7 and Appendix B of the revised manuscript.
>
> ---
> **W2: Lacking the comparison with ShiftAddNet?**
>
> Thank you for pointing out this! We did not include the comparison with ShiftAddNet because the ShiftAddNet paper does not report accuracy on ImageNet. Following your kind suggestion, we supply the comparison with ShiftAddNet to Table 4 of the revised manuscript after confirming with ShiftAddNet authors. As compared to ShiftAddNet, the proposed ShiftAddNAS achieves +9.8% higher accuracy (82.1% vs. 72.3%) and 63% MACs savings (3.7G vs. 10G) under a comparable energy cost (216mJ vs. 223mJ) when measuring on the Eyeriss accelerator.

---

### Official Review · Reviewer_PKFY · 2021-11-02

**Correctness:** 3
**Technical Novelty And Significance:** 2
**Empirical Novelty And Significance:** 3
**Recommendation:** 6
**Confidence:** 3

**Main Review:**

**Strengths**
- The paper is well-motivated, well-written, good quality.
- To my knowledge, it is the first paper to design a hardware-inspired hybrid search space between multiplication-based operators and multiplication-free operators for NAS.
- Clear contribution: this work tackles the problem caused when applying weight-sharing-based or supernet-based search algorithms on the hybrid search space by proposing heterogenous weight sharing.
- Solid empirical experiments including SOTA baseline methods.

**Weaknesses**
- Search algorithm is not new.
- If I read correctly, ShiftAddNet which is one of the most relevant baselines of the proposed work is missing in the experiment section. The reviewer thinks that adding ShiftAddNet in the main Table 4 (or additional analysis section) is needed to show the performance improvement with increasing latency/energy compared with ShiftAddNet.
- The reviewer thinks the cost to train supernet and search cost should be reported.
- Generality issue on multiple hardware devices and energy & latency information are missing in Table 4. Even if this work handles 'hardware-inspired NAS', the target device in the main experiments is only eyeriss. The reviewer thinks that this work should validate the generality of this method in more cases by applying this method on multiple hardwares (at least GPU/CPU/Mobile) and reporting latency/energy saving.
- [Minor] in Table 4, the number of parameters and MACs of the model obtained by this method is not SOTA, yet it is bolded. It would be better to be modified.

**Summary Of The Paper:**

This paper designs a hybrid search space that includes multiplication-based operators and multiplication-free operations to find good trading points between accuracy-efficiency. Further, this work defines the problem when training weight-sharing supernet on the hybrid search space and proposes the heterogenous weight sharing algorithm to address the problem. The proposed method is validated on both NLP and CV tasks, outperforming several competitive baseline methods in terms of accuracy and saving efficiency on latency or energy.

**Summary Of The Review:**

The reviewer thinks that the proposed method is clear enough and practical to obtain neural architecture used for resource-constrained edge devices, yet, some experiments or information that are mentioned above should be added to validate the method.

---

> ### Author Response · Authors · 2021-11-20
> **Response to Reviewer PKFY**
>
> We greatly appreciate your careful review and constructive suggestions. Below are our detailed responses:
>
> ---
> **W1: Search algorithm is not new?**
>
> Yes, you are right and we do not claim the one-shot NAS as a new contribution. As you pointed out,  this is the first paper to design a hardware-inspired hybrid search space and we tackle the problem caused when applying weight-sharing-based or supernet-based search algorithms on the hybrid search space by proposing heterogenous weight sharing, supported with solid empirical experiments including SOTA baseline methods. As shown in Sec. 4.4, our heterogeneous weight sharing strategy leads to a higher ranking correlation and more accurately searched subnets (+1.4% higher accuracy) than the naive weight sharing.
>
> ---
> **W2: Missing the comparison with ShiftAddNet?**
>
> Thank you for pointing out this! We did not include the comparison with ShiftAddNet because the ShiftAddNet paper does not report accuracy on ImageNet. Following your kind suggestion, we supply the comparison with ShiftAddNet to Table 4 of the revised manuscript after confirming with ShiftAddNet authors. As compared to ShiftAddNet, the proposed ShiftAddNAS achieves +9.8% higher accuracy (82.1% vs. 72.3%) and 63% MACs savings (3.7G vs. 10G) under a comparable energy cost (216mJ vs. 223mJ) when measuring on the Eyeriss accelerator.
>
> ---
> **W3: Missing the supernet training cost and search cost?**
>
> Following your kind suggestion, we supply the supernet training cost and search cost for both NLP and CV tasks to the Appendix A of the revised manuscript. For NLP tasks, with one Nvidia V100 GPU, ShiftAddNAS uses on-average 9.3 GPU days (8.5 GPU days for supernet training and 0.8 GPU days for architecture searching) for searching, which is comparable to HAT and 9,821x faster than the Evolved Transformer; For CV tasks, ShiftAddNAS uses on-average 8.9 GPU days (7.7 GPU days for supernet training and 1.2 GPU days for architecture searching) for searching, which is 82% and 11% less than DARTS and BossNAS, respectively.
>
> ---
> **W4: Multiple hardware evaluation except for Eyeriss?**
>
> As mentioned in both AdderNet [1][2] and ShiftAddNet [3], the full potential of hardware-inspired shift and add operations need customizable hardware to support and unfold, otherwise, the latency/energy savings will be proportional to the MAC savings. Following your advice, we further measure the latency/energy using a SOTA mobile FPGA platform (ZYNQ-7 ZC706) through Verilog implementation and simulation of each operator, and find that ShiftAddNAS searched models consistently achieve 17% ~ 85% and 20% ~ 89% latency and energy savings over all baselines in Fig. 6, respectively.
>
> [1] Want et al., AdderNet and its Minimalist Hardware Design for Energy-Efficient Artificial Intelligence, arXiv’21
>
> [2] Li et al., Winograd Algorithm for AdderNet, ICML’21
>
> [3] You et al., ShiftAddNet: A Hardware-Inspired Deep Network, NeurIPS’20
>
> ---
> **W5: Unbold the number of parameters and MACs in Table 4?**
>
> Thanks for pointing this out! We have unbold them in the revised manuscript.

---

### Decision · Program_Chairs · 2022-01-20

**Decision:**

Reject

**Comment:**

This paper develops a hybrid search space consisting of both multiplication-based and multiplication-free operators. It also presents a weight-sharing mechanism for searching in the introduced search space.

Pros:
* A hybrid search space is developed.
* Strong empirical results are reported for both CV and NLP tasks.
* The paper is well written and is easy to follow.

Cons:
* Incremental technical novelty.
* Missing baselines and competing methods.
* Missing information on the search cost.
* Lack of insights into the discovered architectures

The rebuttal has provided most missing information and comparisons, and it has provided additional insights into the searched architectures. However, the reviewers still rate this paper at borderline primarily due to the limited technical novelties. Unfortunately, given these concerns, this submission does not meet the bar for acceptance at ICLR.